# Quantification of dimethyl sulfide (DMS) production in the sea anemone *Aiptasia* sp. to simulate the sea-to-air flux from coral reefs

Filippo Franchini[1*], Michael Steinke[1]

[1]Coral Reef Research Unit, School of Biological Science, University of Essex, Wivenhoe Park, Colchester, CO4 3SQ, United Kingdom

*Correspondence to*: Michael Steinke (msteinke@essex.ac.uk)

**Abstract.** The production of dimethyl sulfide (DMS) is poorly quantified in tropical reef environments but forms an essential process that couples marine and terrestrial sulfur cycles and affects climate. Here we quantified net aqueous DMS production and the concentration of its cellular precursor dimethylsulfoniopropionate (DMSP) in the sea anemone *Aiptasia* sp., a model organism to study coral-related processes. Bleached anemones did not show net DMS production whereas symbiotic anemones produced DMS concentrations (mean ± standard error) of $160.7 \pm 44.22$ nmol $g^{-1}$ dry weight (DW) after 48 h incubation. Symbiotic and bleached individuals showed DMSP concentrations of $32.7 \pm 6.00$ and $0.6 \pm 0.19$ µmol $g^{-1}$ DW, respectively. We applied these findings to a Monte-Carlo simulation to demonstrate that net aqueous DMS production accounts for only 20% of gross aqueous DMS production. Monte-Carlo based estimations of sea–to–air fluxes of gaseous DMS showed that reefs may release up to 25 µmol DMS $m^{-2}$ coral surface area (CSA) $d^{-1}$ into the atmosphere with 40% probability for rates between 0.5 and 1.5 µmol $m^{-2}$ CSA $d^{-1}$. These predictions were in agreement with directly quantified fluxes in previous studies. Conversion to a flux normalised to sea surface area (SSA) (range 0.3 to 17.0 with highest probability for 0.3 to 1.0 µmol DMS $m^{-2}$ SSA $d^{-1}$), suggests that coral reefs emit gaseous DMS at lower rates than the average global oceanic DMS flux of 6.7 µmol $m^{-2}$ SSA $d^{-1}$ (28.1 Tg sulfur per year). The large difference between simulated gross and quantified net aqueous DMS production in corals suggests that the current and future potential for its production in tropical reefs is critically governed by DMS consumption processes. Hence, more research is required to assess the sensitivity of DMS-consumption pathways to ongoing environmental change to address the impact of predicted degradation of coral reefs on DMS production in tropical coastal ecosystems and its impact on future atmospheric DMS concentrations and climate.

## 1 Introduction

The DMSP-catabolite DMS is a biogenic volatile organic compound (BVOC) that provides the dominant natural source of marine sulfur to the atmosphere with a release of 19.6 Tg S per year (Land et al., 2014). This biogenic sulfur affects cloud formation and climate (Vallina and Simó, 2007), and represents the key link in marine and terrestrial sulfur biogeochemical cycling (Bates et al., 1992). However, atmospheric DMS constitutes only a small fraction of the total DMSP and DMS produced in the sea. Less than 20% of dissolved DMSP is directed towards DMS production in planktonic communities (Kiene et al., 2000), and further chemical and biological loss processes including the conversion to dimethyl sulfoxide (DMSO), methanethiol, and formaldehyde by

DMS-oxidising bacteria (Kiene and Bates, 1990; Lidbury et al., 2016), severely limit its availability for sea–to–air transfer, a limiting step for functioning in climate-cooling.

The cnidarian symbiont *Symbiodinium* sp. is a strong producer of DMSP and DMS (Steinke et al., 2011). Hence, the symbiotic sea anemone *Aiptasia* sp. (Van Alstyne et al., 2009) and corals from the Great Barrier Reef (Broadbent and Jones, 2004; Jones and King, 2015) have been found to produce high quantities of DMSP and DMS that fuel the microbial biogeochemistry in coral reefs (Raina et al., 2009). Coral bleaching from the expulsion of *Symbiodinium* endosymbionts occurs regularly as an acclimatisation strategy to monthly and seasonal changes in environmental parameters such as light and temperature. However, climate anomalies can lead to prolonged loss of symbionts and death of the coral (Suggett and Smith, 2011). The principal cause of bleaching is the overproduction of harmful reactive oxygen species (ROS) mostly originating from the photoinhibition of Photosystem II at increased temperature and irradiance (Tchernov et al., 2011), and *Symbiodinium* can provide clade-specific defences to harmful ROS including enhanced protection against UV radiation (Baker, 2003), higher growth (Little et al., 2004), and increased thermal tolerance (Baker et al., 2004). Since DMSP and DMS readily scavenge ROS (Sunda et al., 2002) and algae are known to use DMS to mitigate ROS-induced metabolic damage under sublethal environmental stresses (Archer et al., 2010; Dani and Loreto, 2017), it is possible that they are part of an antioxidant mechanism that leads to the scavenging of ROS and production of DMSO in symbiotic cnidarians (Gardner et al., 2016; Jones and King, 2015).

Tropical sea anemones belonging to the genus *Aiptasia* provide a powerful model organism to investigate the cnidarian host–symbiont relationship in the context of climate change (Baumgarten et al., 2015; Belda-Baillie et al., 2002). In contrast to corals, these anemones can be grown under the presence and absence of their symbionts. This offers unique opportunity to start dissecting the complex interactions between the main DMSP producer (*Symbiodinium* sp.), its host (*Aiptasia*) and the associated microbial community that, taken together, make up the anemone holobiont that releases DMS into the environment. Since information on the sea–to–air flux of DMS and other BVOCs from tropical reefs is scarce (Exton et al., 2014), this study quantified for the first time net aqueous DMS production (net $DMS_{aq}$ production) in *Aiptasia* sp. and used this data together with information on measured DMSP concentration within anemone holobionts ($DMSP_H$) to simulate anemone gross aqueous DMS production (gross $DMS_{aq}$ production) and coral–derived sea–to–air flux of gaseous DMS (net $DMS_g$ flux).

## 2 Methods

### 2.1 Anemone husbandry, bleaching and biomass estimation

The symbiotic tropical sea anemone *Aiptasia* c.f. *pallida* was kept under standard growth conditions in glass aquaria filled with artificial seawater (ASW; 32 g $L^{-1}$ Reef Salt; D-D $H_2Ocean$) inside an incubator (SANYO Versatile Environmental Test Chamber MLR-351) set to 26°C and 12h:12h light/dark cycle at a light intensity of 80 µmol $m^{-2}$ $s^{-1}$. No attempts were made to remove bacteria from the anemones since antibiotic treatment is often detrimental to *Symbiodinium* growth (Yost and Mitchelmore 2009) and we expect the microbial community to be representative of laboratory-grown *Aiptasia*. ASW was changed weekly and the anemones were fed with freshly-hatched brine shrimps (*Artemia salina*, reefphyto) every 2 weeks.

Three months before the start of our measurements, symbiotic anemones were bleached following a cold-shock protocol described in Muscatine et al. (1991). Briefly, they were starved for three weeks, gently removed from their attachment site and transferred to individual 4.92 mL glass vials containing ASW at 26°C. After attachment of the anemones to the glass surface, the water was replaced with cold (4°C) ASW, the vials were closed, kept in the fridge for at least 4 h before replacing the ASW medium and transferring the vials to 26°C. After 1–2 days, anemones were microscopically checked for symbionts using a dissecting microscope and, in case of visually incomplete bleaching, the protocol was repeated. Bleached anemones were kept in darkness but all other growth conditions remained the same.

For biomass estimation, the anemones were anaesthetised in a 1:1 solution of ASW and 0.37M $MgCl_2$, and placed under a dissecting microscope equipped with an eyepiece graticule that was calibrated to the selected magnification. Two oral disk diameters per individual were measured from photographs. Dry and wet weights (DW and WW, respectively) were estimated using the non-linear model for composite treatment proposed earlier (Clayton Jr and Lasker, 1985), and the assumption that the water content in sea anemones is 85% (Brafield and Chapman, 1983).

## 2.2 Experimental design

Before the start of the experiment, bleached and symbiotic anemones were acclimated for 2 months at standard growth conditions in darkness or light, respectively. At the beginning of the experiment, anemones were haphazardly selected for four treatments (n=6 each): Symbiotic light, symbiotic darkness, bleached light and bleached darkness. Samples were incubated for 48 h together with six ASW-filled control vials, before quantifying net DMS production and DMSP concentration.

## 2.3 Quantification of $DMSP_H$ concentration

$DMSP_H$ (i.e. DMSP in anemone holobionts) was indirectly quantified after equimolar hydrolytic conversion to DMS in 3 mL of 0.5M NaOH. DMS was then measured using gas chromatography with flame-photometric detection (GC–FPD) as described earlier (Franchini and Steinke, 2017). Briefly, depending on the amount of DMSP in the specimen, either headspace direct injection of gaseous phase or the more sensitive in vial purging of aqueous phase techniques were used to quantify DMS. For the former technique, 200 µL of headspace were directly injected into the gas chromatograph (GC-2010, Shimadzu, Milton Keynes, UK). For the latter technique, the NaOH in the vials was purged for 6 min with nitrogen (30 mL min$^{-1}$) and this sample gas dried with a Nafion counter flow drier (Permapure MD-050-72F-2, Fluid Controls Limited, Aldermaston, UK) and cryogenically enriched at -150°C using a purpose-built purge-and-trap apparatus, before heating the enriched sample to 90°C and flushing it into the gas chromatograph for quantification. Both techniques were calibrated using DMSP standard solutions (Franchini and Steinke, 2017).

## 2.4 Quantification of holobiont net $DMS_{aq}$ production

To quantify the net production of DMS by the holobiont (net $DMS_{aq}$ production; the release of DMS into the aqueous medium over time), individual anemones were transferred into 3 mL fresh ASW inside 4.92 mL vials and incubated for 48 h. Vials without anemones served as the control. Net $DMS_{aq}$ production was calculated as

the difference in DMS concentration between control vials and vials with anemones after quantification of DMS using the in vial purging of aqueous phase technique (Franchini and Steinke, 2017).

**2.5 Simulating the coral-driven sea–to–air DMS$_g$ flux**

The coral-driven sea–to–air flux of gaseous DMS (net DMS$_g$ flux) was estimated in four steps: (i) simulating the holobiont gross DMS$_{aq}$ production rate using quantified holobiont DMSP concentration, (ii) calculating the ratio ($R$) between measured net and simulated gross DMS$_{aq}$ production, (iii) simulating coral gross DMS$_{aq}$ production rate, and (iv) converting coral gross DMS$_{aq}$ production to coral net DMS$_{aq}$ production (under the assumption that $R$ is similar for anemones and corals) and subsequently to sea–to–air flux using conversion parameters from the literature (Fig. 1; Tables 1 and 2).

Holobiont DMSP concentration from this study was used to simulate the gross DMS$_{aq}$ production rate defined as the total amount of DMS$_{aq}$ produced over time by *Symbiodinium* of clade *i* within the host. Data for cellular DMS production of fee-living *Symbiodinium* from four clades were used as a proxy for gross DMS production (Steinke et al., 2011). The equation describing the holobiont gross DMS$_{aq}$ production rate ($holobiont\ gross_r\ DMS_{aq}$) took the form:

$$holobiont\ gross_r\ DMS_{aq} = \sum_i \left( \frac{DMSP_H \times \frac{N_i}{N_{tot}}}{cDMSP_i} \times cV_i \times cDMS_{aq,i} \right) \quad (1)$$

where $DMSP_H$ is the measured DMSP within the anemone holobiont, $N_i$ is the number of *Symbiodinium* cells of clade *i* (with i= clades A1, A2, A13, B1; see below), and $N_{tot}$ is the total number of cells of different *Symbiodinium* clades (i.e. $\sum_i N_i$). Note that *N* does not reflect the actual number of symbionts within anemones but was arbitrarily set to 100 in order to calculate the proportion of clade *i* among all clades within anemones (setting *N* to $10^3$ or $10^6$ did not change the final outcome of the simulation). This made it possible to generate symbiont communities of different relative compositions during our simulations. Values for $cDMSP_i$ (i.e. cellular DMSP concentration for *Symbiodinium* clade *i*), $cDMS_{aq,i}$ (i.e. cellular DMS$_{aq}$ production rate for *Symbiodinium* clade *i*), and $cV_i$ (cell volume for *Symbiodinium* clade *i*) specific to the free-living *Symbiodinium* clades A1, A2, A13 and B1 were obtained from Steinke et al. (2011) (Table 2; Fig. 1).

The term $DMSP_H \times \frac{N_i}{N_{tot}}$ reflected the contribution of clade *i* to the amount of DMSP within the holobiont. This was divided by $cDMSP_i$ to estimate the number of clade *i* cells per anemone biomass, which was then multiplied by $cV_i$ to obtain the volume occupied by clade *i* per anemone biomass. This biomass-normalized volume was subsequently multiplied by $cDMS_{aq,i}$ to estimate the gross DMS$_{aq}$ production rate per anemone biomass for clade *i*. The sum across all 4 clades yielded the gross DMS$_{aq}$ production rate per anemone biomass.

The fraction of DMS$_{aq}$ released into the water by the anemones was calculated as the ratio ($R$) between the measured net DMS$_{aq}$ production and the simulated gross DMS$_{aq}$ production rate multiplied by the incubation period (i.e. 48 h). Thus, $R$ accounted for the reaction of DMS with ROS and microbial DMS$_{aq}$ consumption pathways mostly related to anemone or coral membrane-associated microorganisms (Fig. 1). The equation for the simulated daily coral-driven sea–to–air flux of gaseous DMS (*net DMS$_g$ flux*) normalised by coral surface area (CSA; µmol m$^{-2}$ d$^{-1}$) took the form:

$$net\ DMS_g\ flux = coral\ gross_r DMS_{aq} \times TW \times R \times P \quad (2)$$

where $coral\ gross_r DMS_{aq}$ is the simulated coral gross $DMS_{aq}$ production rate calculated as in eq. 1 but replacing $DMSP_H$ with $DMSP_C$ (i.e. biomass-normalized DMSP within corals), $TW$ is the coral tissue weight normalized by coral surface area, and $P$ is the percentage of net $DMS_{aq}$ production escaping into the atmosphere (Fig. 1; Table 1). Note that the range of $TW$ was based on values for different coral types (branching, plating, and massive corals) but no efforts were made to explicitly explore different coral types at this stage.

**2.6 Data analysis**

Graphical representations as well as statistical and sensitivity analyses were performed using the R software (R Project for Statistical Computing, version 3.1.1). Datasets for net $DMS_{aq}$ production from light and dark treatments and for comparison between net and gross $DMS_{aq}$ productions were checked for normality and equal variance using a Shapiro-Wilk normality test and Levene's test for homogeneity of variance, respectively. Since all datasets showed non-normal distributions, mono-factorial analyses were performed using the Kruskal-Wallis rank sum test. Treatment and production type were treated as factors (independent variables) with two levels each (light and darkness, and net and gross, respectively). Simulations and sensitivity analysis were performed through the R software package *pse* (Chalom and Knegt Lopez, 2016), following a similar approach to that described in the tutorial by Chalom *et al.* (2013). Briefly, after specifying Equations 1 and 2 and defining all parameters with respective uncertainty ranges and distributions (Tables 1 and 2), we randomly generated 500 values for holobiont $gross_r DMS_{aq}$ and net $DMS_g$ flux through a Monte-Carlo simulation using the LHS (Latin Hypercube Sampling for uncertainty and sensitivity analyses) function within the *pse* package. This function feeds the simulation framework formed by Equations 1 and 2 with random values for each parameter within the specified ranges. Resulting simulation outcomes were collected and used to generate probability distribution plots. Finally, using the LHS function, partial rank correlation coefficients (prcc) were calculated, which indicate the influence of a parameter on the simulation outcome (with 1 = maximum positive influence and -1 = maximum negative influence; 0 = no influence on simulation outcome). These coefficients were used to assess the response (sensitivity) of our simulation framework to variations in each variable.

**3 Results and Discussion**

**3.1 Symbionts are the main source of DMSP and DMS in *Aiptasia***

Symbionts were the main source of DMSP and our data for symbiotic and bleached anemones are in general agreement with the earlier findings (Table 3) (Van Alstyne et al., 2009; Yancey et al., 2010). However, using the more sensitive in vial purging method compared to the headspace sampling performed by Van Alstyne *et al.* (2009), bleached anemones kept in darkness for 2 months showed an average DMSP concentration of $0.6 \pm 0.19$ $\mu mol\ g^{-1}$ DW (n=6). Additional microscopic observation revealed small clusters of symbiont cells within *Aiptasia* tentacles suggesting that bleaching was incomplete, hence, individuals were not aposymbiotic. Whether aposymbiotic anemones produce DMSP as demonstrated for corals (Raina et al., 2013) requires further investigation.

We quantified for the first time the net $DMS_{aq}$ production in *Aiptasia* and demonstrated that the symbiont is the main source of DMS (Fig. 2a). Bleached individuals showed net $DMS_{aq}$ production above the limit of detection but below the limit of quantification at $1.2 \pm 0.62$ nM which is equivalent to a production rate of 3.6 pmol DMS in 3 mL over a 48 h incubation.

## 3.2 Effect of light on DMS production

Net $DMS_{aq}$ production in dark was the same as in light treatments (Fig. 2a). Although light has been shown to affect the cycling of DMS (Galí et al., 2013; Toole and Siegel, 2004), our results indicate that acclimated symbiotic *Aiptasia* produced 52 to 332 nmol DMS $g^{-1}$ DW (mean = $160.7 \pm 44.22$ nmol $g^{-1}$ DW; n = 6) over a 48h incubation period with no significant difference between the light and dark treatments (P=0.42; Fig. 2a). Various DMS removal processes affect the amount of DMS that could be detected in the water surrounding the anemones and our measurements represent net $DMS_{aq}$ production rates. Microbial consumption of DMS is concentration dependent and affected by the microbial diversity and presence of DMS-consuming bacteria (Schäfer et al.2010). Although the microbial community associated with the holobiont surface is probably conservative, exchanging the seawater medium (ASW) at the start of our incubations likely resulted in a low abundance of free-living bacteria in comparison to the natural setting. Furthermore, consumption of DMS may be sensitive to light since photosynthetically derived $O_2$ could stimulate heterotrophic respiration of DMS. In fact, the activity and population size of DMS-oxidising bacteria are higher during oxic/light than during anoxic/dark conditions (Jonkers et al., 2000). Moreover, light is expected to increase ROS that could oxidise DMS and produce DMSO, hence, contributes to DMS consumption (Fig. 1). This scenario suggests that DMS consumption could be higher during the day than at night, and that, as a consequence, net production should show the opposite pattern. However, light can also result in an increase of DMS in phytoplankton suggesting a direct link between acute photo-oxidative stress and DMSP synthesis but the physiological basis for this is unclear (Archer et al., 2010).

## 3.3 From anemones to corals: Net vs. gross $DMS_{aq}$ production and net DMS flux

Using our measurements of DMSP concentration and net DMS production in anemones to extrapolate to coral reef environments provides an initial route to assess overall DMS production in tropical coastal ecosystems where DMS and DMSP data coverage is relatively poor. To highlight the basis of our approach and discuss possible limitations we will first examine five major assumptions in our approach:

(i) Endosymbionts are the main DMSP and DMS producers within anemones and corals. Juvenile corals of the high DMSP-producing genus *Acropora* that were aposymbiotic (free from *Symbiodinium* symbionts) showed a low level of DMSP production. This suggests that corals and possibly other cnidarians can be a cryptic source of DMSP that is not generated by their endosymbiotic partner. Our bleached anemones were not aposymbiotic in our experiment and showed low DMSP concentrations with DMS production below the level of quantification. This confirms a previous study that could not detect DMSP in aposymbiotic anemones (Table 3; Van Alstyne et al., 2009) and supports our assumption that the endosymbionts are the main producers of DMSP and DMS.

(ii) There is no difference in cellular DMSP content (*cDMSP*) and $DMS_{aq}$ production rate (*cDMS$_{aq}$*) between free-living *Symbiodinium* cells and those living symbiotically. Symbionts in corals contained about 10 to 300

fmol DMSP cell$^{-1}$ (Yost and Mitchelmore 2010), while Borell et al. (2014; 2016) reported concentrations ranging from about 20 to 50 fmol DMSP cell$^{-1}$. These values for both corals and anemones were similar to the 4 free-living *Symbiodinium* clades investigated by Steinke et al. (2011) (39.3 to 126.8 fmol DMSP cell$^{-1}$; Table 2) and suggest that free-living and endosymbiotic *Symbiodinium* likely produce similar amounts of cellular DMSP. As far as we are aware, Steinke et al. (2011) present the only data for *cDMS$_{aq}$* in *Symbiodinium* so that this assumption cannot be tested against other published information.

(iii) DMSP and DMS characteristics in *Symbiodinium* clades A1, A2, A13, and B1 are representative of other symbiont types. Symbiont community composition varies depending on the geographic region. In the Red Sea and in the Sea of Oman clade A1 was found to be one of the most abundant (Ziegler et al. 2017), while clade B1 was found to be abundant in Caribbean reef-building coral *Orbicella faveolata* (Kemp et al. 2015). Both clades seemed to play a minor role in the Indo-Pacific (Yang et al. 2012). Although having information about DMS and DMSP characteristics for more clades might improve simulation accuracy, such values seem to play a minor role in shaping the DMS sea–to–air flux (see below and Figure 3).

(iv) The ratio between net and gross DMS$_{aq}$ production calculated for anemones also applies to corals, an assumption that is currently impossible to test due to the lack of published information.

(v) Finally, light intensity does not significantly affect *cDMS$_{aq}$*. Although light conditions in the experiment conducted by Steinke *et al.* (2011) (350 μmol m$^{-2}$ s$^{-1}$, 14h:10h light/dark cycle) were different from those adopted here, the evidence that net DMS$_{aq}$ production was independent of light intensity (see Sect. 3.2) is in support of our assumption.

*Aiptasia* is an accepted model to investigate the cnidarian host–symbiont relationship (Baumgarten et al., 2015; Belda-Baillie et al., 2002). However, the microbial communities on the surface of corals and anemones may differ, leading to potential differences in DMS biogeochemistry. Corals produce vortical ciliary flows that may not only limit the attachment of pathogens to the host surface, but also prevent accumulation of oxygen that could inhibit the photosynthetic activity of their endosymbionts (Shapiro et al. 2014). Whether those ciliary flows are also present in anemones has to be investigated.

The adopted simulation framework suggests that gross DMS production of ~1 μmol g$^{-1}$ over 48 h is up to 7 times higher than the net production of ~0.15 μmol g$^{-1}$ (P < 0.001) (Fig. 2b). Additionally, the percentage of the gross production escaping into the water surrounding the anemones ranged from 1 to 120%. A percentage of gross DMS production escaping into the atmosphere greater than 100% occurs when the simulated net production exceeds the gross production. This is due to the random sampling of high net-production values (Table 1), and to the calculation of low gross production within the simulation framework. Low gross production arises when a particular combination of parameter values are inserted into Equation 1. For example, a low symbiont population resulting in low DMSP$_H$, combined with a population of low DMS-producing *Symbiodinium* clades with small cell volume (e.g. *Symbiodinium* clade B1) could result in this output from the simulation framework. Highest probabilities of 60% were found for net/gross production ratios of 5 to 30% (Fig. 2c). It is proposed that the remaining 70 to 95% reacts with ROS or is consumed in other ways by the host and surface-associated microorganisms (Fig. 1). It has been demonstrated that the coral host not only controls the population size of various *Symbiodinium* clades inside the symbiosomes (Kemp et al., 2014), but it also actively modifies the microenvironment on their surface (Barott et al., 2015), both with consequences for DMSP concentration and DMS production. Furthermore, DMS production is significantly different between the

*Symbiodinium* clades (Table 2) so that the relative abundance of clade A1 affected coral–driven sea–to–air DMS fluxes (see $N_{A1}$ in Figure 3), which ranged from 0 to 25 µmol m$^{-2}$ d$^{-1}$ with 40% probability for values between 0.5 and 1.5 µmol m$^{-2}$ CSA d$^{-1}$ (Fig. 2d). The other clades had minor influence on sea–to–air DMS fluxes, because even if corals accommodate high DMS-producing endosymbionts leading to high gross DMS$_{aq}$ production rates, the amount of DMS emitted into the atmosphere is more strongly affected by physico–chemical variables including temperature (affects DMS solubility) and wind speed (drives sea–to–air transfer), and depends more critically on net DMS$_{aq}$ production that is the result of several DMS-consumption processes (Fig.1; Fig. 3). Simulation parameters that highly influenced the DMS flux were the percentage of aqueous DMS escaping into the atmosphere (P), the coral tissue weight normalised by coral surface area (TW), coral DMSP (DMSP$_C$), holobiont DMSP (DMSP$_H$), and anemone net DMS$_{aq}$ production (Fig. 3). This is not surprising since P shapes the amount of dissolved DMS escaping into the atmosphere, TW is a proxy of reef structural complexity and the higher this is, the larger is the surface area per unit of biomass available to accommodate DMS-producing symbionts, and DMSP$_C$ and DMSP$_H$ are proportional to the total number of symbionts. Since DMSP$_H$ is used to simulate anemone gross DMS$_{aq}$ production and subsequently to estimate R, higher DMSP$_H$ will decrease R, resulting in decreased DMS flux (Fig. 3). Finally, the larger the pool of DMS dissolved in the water (net DMS$_{aq}$ production) the higher the chance that DMS will escape into the atmosphere.

The range of sea–to–air DMS fluxes obtained from our simulation is in good agreement with earlier measurements on *Acropora intermedia*, a dominant staghorn coral in the Indo-Pacific region, which generated a sea–to–air flux of 0.55 to 1.13 µmol m$^{-2}$ CSA d$^{-1}$ (Fischer and Jones, 2012). Converting CSA-normalised fluxes into fluxes normalised to sea surface area (SSA) requires information on coral cover and reef rugosity, i.e. the unit-less ratio between the reef real surface area and its projected area with a ratio of 1 indicating a flat reef while rugosity values >1 indicate increasing structural complexity. Assuming a coral cover of 22% in the Indo-Pacific (Bruno and Selig, 2007) and an average rugosity of 3 (Storlazzi et al., 2016), we can calculate a maximum flux of about 17 µmol DMS m$^{-2}$ SSA d$^{-1}$ with highest probabilities for fluxes ranging from 0.3 to 1 µmol DMS m$^{-2}$ SSA d$^{-1}$. This flux is in good agreement with modelled fluxes based on continuous DMS measurements during the wet and dry seasons at Heron Island in the southern Great Barrier Reef that show coral-derived DMS fluxes of 0.2 µmol DMS m$^{-2}$ SSA d$^{-1}$ (Swan et al., 2017). Taken together, this suggests that coral reefs likely continuously emit DMS at lower rates than the short-lived DMS 'hotspots' of phytoplankton blooms in the North Atlantic (20.69 to 26.93 µmol m$^{-2}$ SSA d$^{-1}$; Holligan et al., 1993) or at high latitudes (21.87 µmol m$^{-2}$ SSA d$^{-1}$; Levasseur et al., 1994). Furthermore, our estimated sea-to-air flux from coral reefs is also often lower than the global oceanic flux that is calculated at 4.6 µmol m$^{-2}$ SSA d$^{-1}$ (equivalent to 19.6 Tg S y$^{-1}$ in Land et al., 2014) and is in agreement with earlier findings that suggest coral environments enhance the dominant oceanic DMS flux by just 4% during the wet season and 14% during the dry season (Swan et al., 2017). While our calculated fluxes refer to fully submersed reefs, it is important to note that tidally-exposed corals such as the strongly DMS-producing *Acropora* spp. (3,000–11,000 µmol DMS m$^{-2}$ SSA d$^{-1}$) may provide substantial short 'bursts' of DMS to the atmosphere that last for several minutes during and after periods of aerial exposure (Hopkins et al., 2016). These bursts can be further enhanced when corals experience hypoosmotic stress from rain (Swan et al., 2017).

Our study suggests that net DMS$_{aq}$ production and the resulting sea–to–air flux from coral reefs are under strong control of DMS-consumption pathways. Furthermore, DMS and its massively abundant precursor DMSP

(Broadbent and Jones, 2004) are likely key metabolites that significantly fuel microbial activity in tropical coastal ecosystems (Raina et al., 2009). Hence, predicting future DMS concentration in the tropical atmosphere and its effect on climate requires an assessment of the sensitivity of DMS-consumption processes in reefs under environmental change.

## 4 Data availability

The datasets supporting this article will be made publicly available upon manuscript acceptance.

## 5 Author contribution

F. Franchini and M. Steinke conceived and designed the study, interpreted the data and wrote the manuscript. F. Franchini performed the experiments, and collected and analysed the data. Both authors gave final approval for publication.

## 6 Competing interests

The authors declare that they have no conflict of interest.

## 7 Acknowledgments

The authors thank Sue Corbett, Tania Cresswell-Maynard and John Green for technical assistance.

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

**Table 1:** Parameters used for the simulation. DMS, dimethylsulfide; DMSP, dimethylsulfoniopropionate; DW, dry weight; N/A, not applicable. Note that the simulation was unaffected when setting N (the arbitrary number of clade-specific *Symbiodinium* cells) to a maximum value of 1000.

| Parameter | Description | Unit | min | max | Source |
|---|---|---|---|---|---|
| | | | \multicolumn Range | | |
| $DMSP_H$ | Biomass-normalised DMSP within the anemone holobiont | $\mu$mol g$^{-1}$ DW | 15.09 | 51.82 | This study |
| net $DMS_{aq}$ | Biomass-normalised net aqueous DMS production | nmol g$^{-1}$ DW in 48h | 52.15 | 332.25 | This study |
| TW | Coral tissue weight normalised by coral surface area (CSA) | mg DW cm$^{-2}$ | 2.58 | 11.51 | Thornhill et al. (2013) |
| $DMSP_C$ | Biomass-normalised DMSP within corals | $\mu$mol g$^{-1}$ DW | 52.36 | 84.00 | Yancey et al. (2010) |
| $N_{A1, A2, A13, B1}$ | Arbitrary number of clade-specific *Symbiodinium* cells | N/A | 0 | 100 | – |
| P | Percentage of aqueous DMS escaping into the atmosphere | % | 1 | 20 | Bates et al. (1994) |

**Table 2:** Parameters for cellular DMSP concentration (cDMSP), cell volume (cV) and cellular net $DMS_{aq}$ production rate ($cDMS_{aq}$) in four clades of *Symbiodinium* sp. (data from Steinke et al.2011).

| *Symbiodinium* clade | cDMSP (fmol $cell^{-1}$) | cV ($\mu m^3$ $cell^{-1}$) | $cDMS_{aq}$ (mmol $L^{-1}$ cV $h^{-1}$) |
|---|---|---|---|
| A1 | $98.0 \pm 4.18$ | $415 \pm 9.5$ | $0.32 \pm 0.112$ |
| A2 | $126.8 \pm 8.59$ | $763 \pm 24.2$ | $0.06 \pm 0.018$ |
| A13 | $85.6 \pm 22.03$ | $419 \pm 34.5$ | $0.10 \pm 0.029$ |
| B1 | $39.3 \pm 2.33$ | $237 \pm 19.7$ | $0.04 \pm 0.025$ |

**Table 3:** Biomass-normalised DMSP within symbiotic or bleached anemones (mean ± se) in this and two previous studies. Sample size (n) in parentheses. DMSP, dimethylsulfoniopropionate; DW, dry weight; ND, not detectable; NI, not investigated.

| *Aiptasia* Species | DMSP ($\mu$mol g$^{-1}$ DW) | | Source |
| | Symbiotic | Bleached | |
|---|---|---|---|
| *A. pallida* | 22.7 ± 8.00 (2) | ND (3) | Van Alstyne et al. (2009) |
| *A. puchella* | 54.7 ± 15.20 (3) | NI | Yancey et al. (2010) |
| *A.* cf. *pallida* | 32.7 ± 6.00 (6) | 0.6 ± 0.19 (6) | This Study |

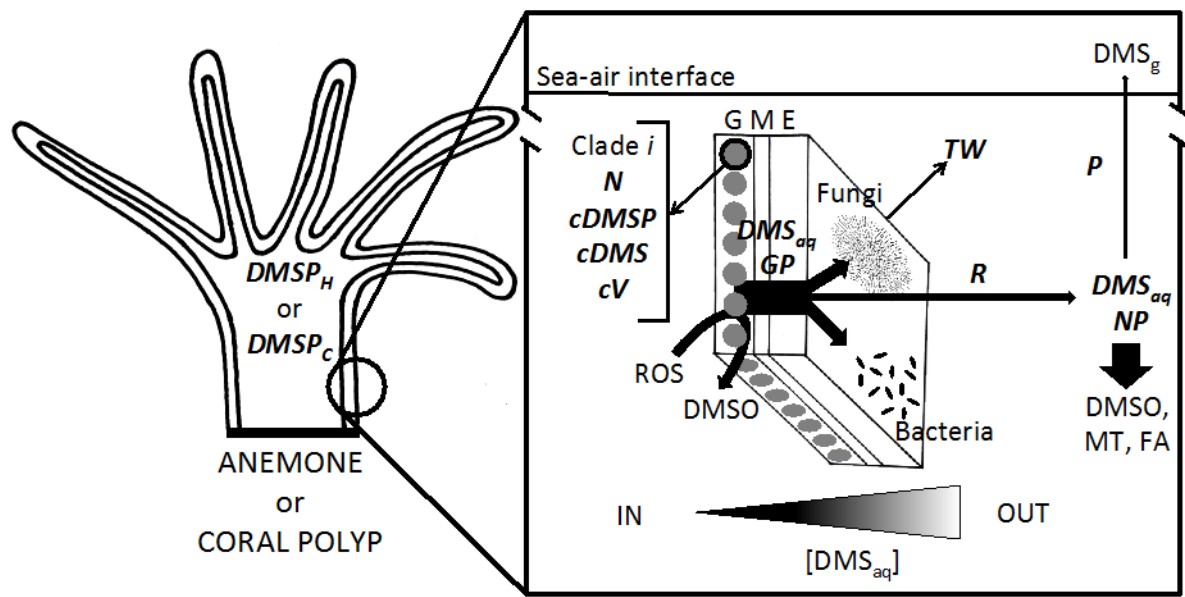

**Figure 1:** Magnification of a coral polyp and its cell layers with particular emphasis on the pathway of DMS from its production by endosymbionts (grey circles) to its release to the atmosphere. Note that symbols in bold italics describe the parameters fed into the simulation framework. The host with a particular tissue weight (TW) accommodates a number $N$ of cells for *Symbiodinium* clade $i$ containing DMSP (cDMSP, cellular DMSP), producing and releasing DMS (cDMS, cellular DMS production rate), and having a particular volume (cV, cellular volume). All cells of all clades $i$ form the total DMSP concentration within the anemone holobiont or corals ($DMSP_H$ and $DMSP_C$, respectively). Measured net $DMS_{aq}$ production ($DMS_{aq}$ NP) is a fraction R of the gross $DMS_{aq}$ production ($DMS_{aq}$ GP). The remaining DMS (i.e. 1-R) is available to scavenge reactive oxygen species (ROS) and/or is consumed by surface-associated microbes. Once dissolved, a fraction P of the net $DMS_{aq}$ production escapes into the atmosphere, while most of it is biologically transformed by free-living bacteria in the water column to, for example, DMSO, methanethiol (MT) and formaldehyde (FA). DMS, dimethylsulfide; $DMS_g$, gaseous DMS; $DMS_{aq}$, aqueous DMS; DMSO, dimethyl sulfoxide; G, gastrodermis; M, mesoglea; E, epidermis.

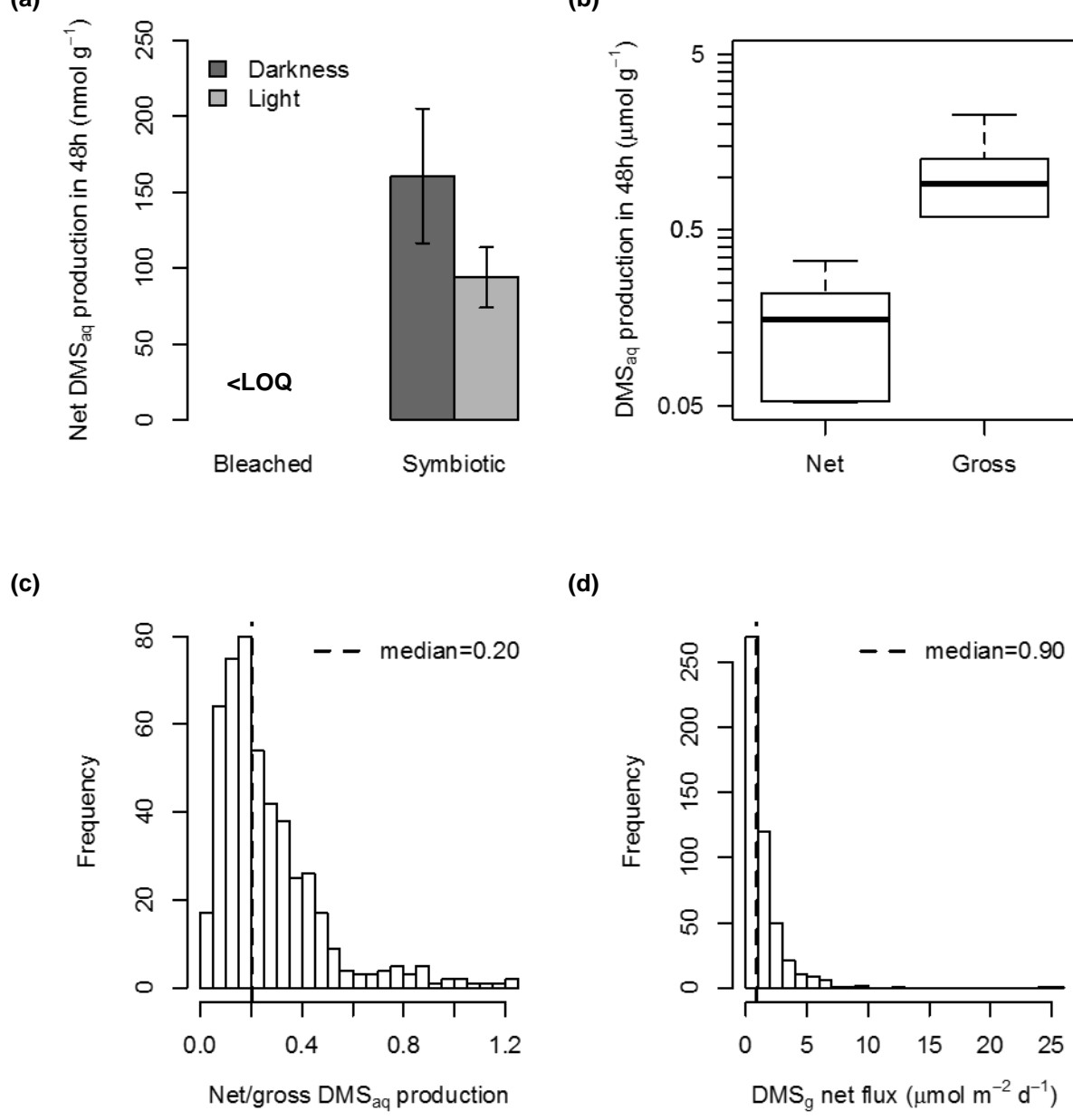

**Figure 2: (a)** Biomass-normalised (dry weight) net DMS production (mean ± se) for symbiotic and bleached anemones during light and dark treatments (n=6). **(b)** Boxplot showing the difference (P < 0.001) between the biomass-normalised (dry weight) observed net $DMS_{aq}$ production (n=6) and the simulated gross $DMS_{aq}$ production after 500 simulations for symbiotic anemones. Boxes show first and third quartile ranges, thick lines indicate median values, and error bars the range of data. Please note the logarithmic scale along the y-axis. **(c)** Distribution of the net/gross production ratio after 500 simulations. **(d)** Distribution for coral-driven daily net $DMS_g$ flux into the atmosphere normalised by coral surface area after 500 simulations. LOQ, limit of quantification; $DMS_{aq}$, aqueous dimethyl sulfide; $DMS_g$, gaseous dimethyl sulfide.

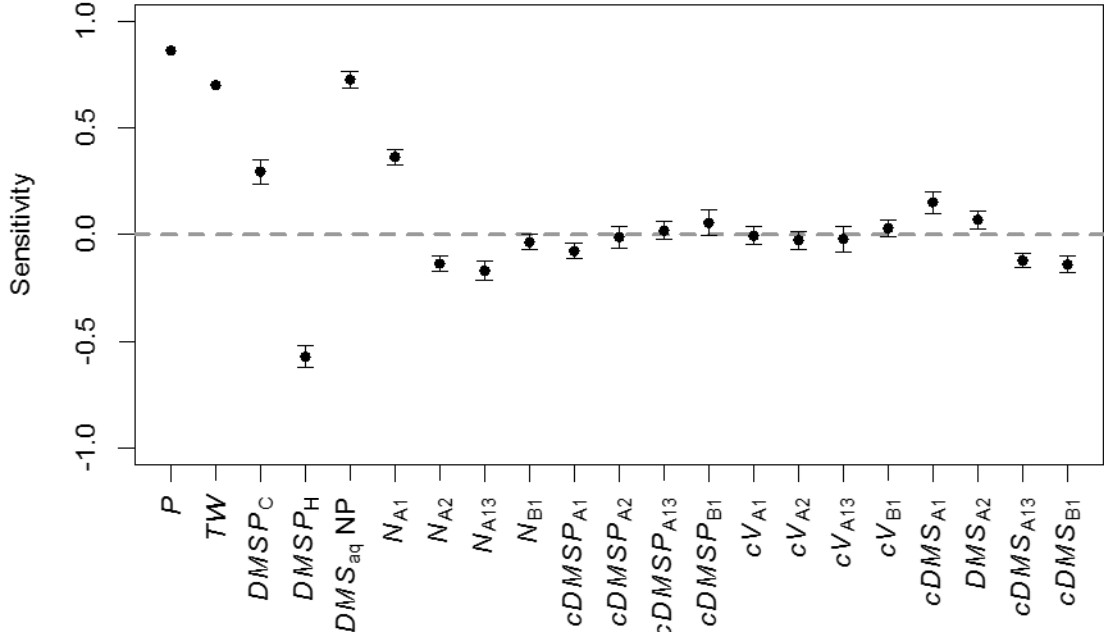

**Figure 3:** Sensitivity of the variables fed into the simulation framework. Values close to 0 have less influence on the simulation than those departing from 0. Error bars show standard error. Where error bars are invisible they are smaller than the symbol size. $DMS_{aq}$ and $DMS_g$, aqueous and gaseous dimethyl sulfide; P, percentage of $DMS_{aq}$ escaping into the atmosphere; TW, coral tissue weight normalised by coral surface area; $DMSP_C$, dimethylsulfoniopropionate within corals; $DMSP_H$, dimethylsulfoniopropionate in holobionts; $DMS_{aq}$ NP, net aqueous DMS production; N, number of cells for *Symbiodinium* clades A1, A2, A13, and B1; cDMSP, cellular DMSP for *Symbiodinium* clades A1, A2, A13, and B1; cV; cell volume for *Symbiodinium* clades A1, A2, A13, and B1; cDMS, cellular $DMS_g$ production rate for *Symbiodinium* clades A1, A2, A13, and B1.