# Peer review of "Quantification of dimethyl sulfide (DMS) production in the sea anemone *Aiptasia* sp. to simulate the sea-to-air flux from coral reefs"

_Biogeosciences, 2017_

## Referee Comment (RC1) · Anonymous Referee #1 · 15 Mar 2017

Review of: "Quantification of DMS production in the sea anemone Aiptasia sp. to simulate the sea-to-air flux from coral reefs" by F. Franchini and Michael Steinke

General: The authors use a sea anemone as a model organism to study DMS flux from coral reefs. There are major deficiencies in this approach and I cannot recommend this manuscript for publication. If anything the results are very preliminary and a gross approximation of DMS flux from coral reefs. This is only superficially acknowledged. Using artificial seawater and cold shock to 4oC to compare bleached and unbleached samples is not realistic. Generally only a 2oC shock above or below ambient seawater

16000

[Figure]

temperatures should be used to stress a coral and would be comparable to studies by Fischer and Jones (2012). No measurements seem to be made on the actual symbiodinium concentrations in samples and results are expressed per gram. Conversion to surface areas should be shown in a table and compared with other available data so that good comparisons can be made. The authors should discuss in length two other important papers that have made good measurements and assessments of DMS flux from coral reefs. These are:

Hopkins, F.E., Bell, T.G., Yang, M., Suggett, D.J. and Steinke, M. (2016) Air exposure of coral is a significant source of dimethylsulphide (DMS) to the atmosphere. Scientific Reports, 6:36031,doi:1038/srep36031.

Swan, H.B., Jones, G.B., Deschaseaux, E.S.M and Eyre, B.D. (2017) Coral reef origins of atmospheric dimethylsulfide at Heron Island, southern Great Barrier Reef, Australia. Biogeosciences, 14, 1-11. Doi 10.5194/bg-14-1-2017

DMS flux can be estimated by both atmospheric and seawater measurements of DMS and the two papers above have shown that corals emit DMS directly to the atmosphere. The submitted paper makes no mention of this in their article. Their measurements from a sea anemone are therefore a gross underestimate. This is not helped by arbitrarily estimating the number of clade types in the anemone and not measuring them in the anemone. Different clades of zooxanthellae contain different levels of DMSP and produce variable levels of DMS. What data is available and published on DMS and DMSP production from coral reefs and discrete corals (e.g. Acropora-the most abundant coral in the Indo-Pacific) is not used or quoted (see Jones et al. (2007); Jones and King (2015).

Scientific significance: Does the manuscript represent a substantial contribution to scientific progress within the scope of Biogeosciences (substantial new concepts, ideas, methods, or data)? Score 4

Scientific quality: Are the scientific approach and applied methods valid? Are the results discussed in an appropriate and balanced way (consideration of related work, including appropriate references)? Score 4

Presentation quality: Are the scientific results and conclusions presented in a clear, concise, and well-structured way (number and quality of figures/tables, appropriate use of English language)? Score 4

---

## Referee Comment (RC2) · Anonymous Referee #2 · 30 Mar 2017

Review of the paper "Quantification of dimethyl sulfide (DMS) production in the sea anemone Aiptasia sp. to simulate the sea-to-air flux from coral reefs" by Filippo Franchini and Micheal Steinke

The paper presented net DMS production and DMSP concentrations in cultures of 48h incubated sea anemones Aiptasia sp. with and without its symbiont Symbiodinium. These data together with literature values were used to estimate the gross DMS production within the anemones and the potential amount of anemone derived DMS emitted to the atmosphere.

This study presents an interesting aspect of the role and influence of sea anemones for the biogeochemical pathways of DMS and DMSP in coral reefs. It shows that even when the production of anemones inside of the polyp is relatively high most of the DMS and DMSP is rapidly consumed and degraded due to microbial activities surrounding the anemones showing again the importance of these sulfur species for the microbial world. Additionally, this study showed that the amount of anemone/coral reef derived DMS for atmospheric processes might be less important than it was thought before suggesting coral reefs as less important hot spots compare with phytoplankton spring blooms in boreal regions. However, the method part of the paper is difficult to understand due to very short descriptions that missing some important details resulting in confusion of the reader. Thus, I suggest publishing this paper after major revision.

Major comments The reader gets easily confused by the different terms "net DMS production" and "DMS gross production" and which of the terms are measured or calculated/estimated. Figure 2 was very helpful to understand but it is mentioned only in the last section of the paper. Please define/specify in your method parts the different terms in one to two sentences and make clear how you determined it.

The anemone gross DMS production calculation is confusing and difficult to understand when it is explained together with the DMS flux calculation in one equation. For a better understanding please explain first the gross DMS production separately and give more information about the different parameter you used in the equation. It is not completely clear why you chose certain parameters. For instance, why you used DMSP from Yancey et al. 2010 when you have directly measured DMSP and biomass in your incubations? Why you chose for NÂňA1, NA2 and so on cell number maximum of 100? Is this a reasonable amount for anemone symbionts in your cultures? And please give more information about your previous study Steinke et al. 2011 regarding DMS and Symbiodinium you refer to in this study. How did you determined TW? And why is P between 0 and 20 % reasonable for your experiment. Why is the equation for gross DMSaq in anemone the same as the coral gross DMS-production equation (p 3,

[Figure]

L 35)? Did you replace the TW for corals in this equation with the TW of the anemone? Why are the assumptions on p4 L1-7 are reasonable. Please justify. Have you tested it?

In your experiment, anemones were the organisms of interest, but you talked a lot about corals and coral surface area, so the reader gets confused if you want to show the impact of anemones or corals. You also said "Using our measurements of DMSP concentration and DMS production in anemones to extrapolate to coral reef environments has its limitations…" (p5, L16). Furthermore, on P5 L28: You said that you "normalized to CSA". How did you normalized? Did you assume that anemone coverage in coral reefs was 100% or you assumed that corals and anemones produce similar amounts of DMS so that the composition of the coral reefs (corals or anemones) didn't matter? Please justify why you can compare anemones and corals and why you can use anemone driven DMS to interpret the amount of DMS produced/released from coral reefs in general. Please say also something about the limitation of this comparison.

In your equation and your Fig 1c, please explain shortly the meaning of the term net DMSaq/gross DMSaq. Does the term say something about the amount of consumed DMS?

The section 2.5 "Data analysis" is very difficult to understand. It needs more details about why and what you were doing with your data. What do you want to say in the first sentence (p4 L10)? Please reword it. Is the mono-factorial analysis well known? Can you shortly say what that mean? What is the R package pse doing, why you used it? The references you gave are very complicated and detailed. It would be great when you give a more general information in your paper. Please, give also a short and general explanation about Monte-Carlo and why you applied it. In the last sentence of section 2.5 (p4 L20-22) is not clear what you have done. Please give more information how you determined the sensitivity of the variables.

Why you didn't determine the net DMSP production? Is this term not interesting?

Minor comments Abstract P1 L10: Please delete the part with the gas chromatograph. I suggest "Here we quantified the net DMS production and the concentration of its cellular precursor dimethylsulfoniopropionate (DMSP) in the cultured sea anemone Aiptasia sp., . . ."

Please show only one number after the decimal place in the abstract, e.g. 44.2 instead of 44.22 (p1 L13) and 6 instead of 6.00.

P1 L15: This sentence is very confusing. You say that you simulated the DMS flux and than you present the results of the gross DMS production. I suggest "We applied these findings to a Monte-Carlo simulation to demonstrate that net aqueous DMS production accounts for only 0.5 – 2% of gross aqueous DMS production. Monte Carlo based estimations of DMS fluxes into the atmosphere showed that reefs may release up to . . ."

Maybe you can write also a discussion sentence about the DMS flux results in the abstract as you have done for DMS gross production.

Section 3.2 You discussed in this section that DMS removal processes under light conditions are faster compared to dark conditions mainly due to microbial consumption. However, in your incubation experiment you didn't see lower DMS concentration in the light treatments compare to the dark treatments. Maybe you should consider and discuss that your incubation experiments didn't contain the microbial diversity as natural environments have. You used artificial seawater (axenic?) for the incubation, thus you might miss important DMS consuming microbes in your experiments resulting in similar DMS concentrations in dark and light treatments.

P6 L 1: "an average rugosity of 3". Can you say what that means? Is 3 much rugosity or only a little bit? Has rugosity a unit?

Fig.1 d: please add the different variables in the figure or color code the dots. It is not clear which point presents which variable in the sensitivity plot. Maybe you can say a

little bit about what the different sensitivity numbers mean in the plot, such as "variables close to 0 have less influence on the simulation than variables lower/higher than 0" or something similar.

---

## Author Comment (AC1) · 26 Jun 2017

Review of: "Quantification of DMS production in the sea anemone Aiptasia sp. to simulate the sea-to-air flux from coral reefs" by F. Franchini and Michael Steinke.

General: The authors use a sea anemone as a model organism to study DMS flux from coral reefs. There are major deficiencies in this approach and I cannot recommend this manuscript for publication. If anything the results are very preliminary and a gross approximation of DMS flux from coral reefs. This is only superficially acknowledged.

Author response (AR) 1: We thank referee 1 for their comments and can provide re-assurance that our results are based on a suitable experimental design where none of the measured data are of preliminary nature. The simulation is based on a series of assumptions that we have clarified in the Results and Discussion section of the revised version. Since information on DMS cycling is severely limited for tropical reefs, we used our model simulation to estimate the flux of DMS from corals and the outcome of our simulation is in excellent agreement with the very few data from previous studies that quantified DMS flux from coral directly (e.g. Fischer and Jones 2012) and with cal-culated fluxes based on continuous atmospheric DMS measurements at Heron Island (Swan et al., 2017). We added information on the study by Swan et al. (2017) and highlighted the limitations of our study in the revised section 3.3.

Using artificial seawater and cold shock to 4oC to compare bleached and unbleached samples is not realistic. Generally only a 2oC shock above or below ambient seawater temperatures should be used to stress a coral and would be comparable to studies by Fischer and Jones (2012).

AR2: We did not use acute cold shock in any of our experiments. We merely used a widely accepted cold-chock protocol for anemones to bleach Aiptasia and reduce the number of endosymbionts to compare the production of DMS between symbiotic and bleached individuals. After the cold shock, there was a period of 3 months where the bleached and non-bleached Aiptasia were acclimated to our experimental conditions.

No measurements seem to be made on the actual Symbiodinium concentrations in samples and results are expressed per gram.

AR3: Our model simulation does not require data on Symbiodinium concentrations but uses measurements of the holobiont DMSP concentration (DMSPH ; a proxy for Symbiodinium concentration in the anemone holobiont) and net DMS production rate (net DMSaq). Other required information was taken from the literature (see Table 1). Following the conventions in previously published studies (Van Alstyne et al. 2009;

[Figure]

Yancey et al. 2010), we expressed DMSP data in units of $\mu$mol g-1 DW (Table 3).

Conversion to surface areas should be shown in a table and compared with other available data so that good comparisons can be made.

AR4: All key data on fluxes normalised to coral surface area and sea surface area in our dataset are clearly presented in the text and compared with data in the literature (Fisher and Jones, 2012; Swan et al., 2017). We also discuss information on the global DMS flux estimates (Lana et al., 2012) and from measurements in the North Atlantic and high latitudes (Holligan et al., 1993; Levasseur et al., 1994). We do not feel that the manuscript would benefit from including a table with this information.

The authors should discuss in length two other important papers that have made good measurements and assessments of DMS flux from coral reefs. These are:

Hopkins, F.E., Bell, T.G., Yang, M., Suggett, D.J. and Steinke, M. (2016) Air exposure of coral is a significant source of dimethylsulphide (DMS) to the atmosphere. Scientific Reports, 6:36031,doi:1038/srep36031.

Swan, H.B., Jones, G.B., Deschaseaux, E.S.M and Eyre, B.D. (2017) Coral reef origins of atmospheric dimethylsulfide at Heron Island, southern Great Barrier Reef, Australia. Biogeosciences, 14, 1-11. Doi 10.5194/bg-14-1-2017

DMS flux can be estimated by both atmospheric and seawater measurements of DMS and the two papers above have shown that corals emit DMS directly to the atmosphere. The submitted paper makes no mention of this in their article.

AR5: Information from the publication by Hopkins et al. (2016) was included in our initial submission of the manuscript. We provide more information on their findings in the revised Results and Discussion section to highlight the importance of short 'bursts' of DMS during periods of aerial exposure. The paper by Swan et al. (2017) was not included in our initial submission (it was published one month before our initial submission). We apologise for this oversight and have included a discussion of their

relevant key findings in the revised version.

Their measurements from a sea anemone are therefore a gross underestimate. This is not helped by arbitrarily estimating the number of clade types in the anemone and not measuring them in the anemone. Different clades of zooxanthellae contain different levels of DMSP and produce variable levels of DMS. What data is available and published on DMS and DMSP production from coral reefs and discrete corals (e.g. Acropora-the most abundant coral in the Indo-Pacific) is not used or quoted (see Jones et al. (2007); Jones and King (2015).

AR6: Results from our flux simulation are in excellent agreement with the very few published datasets that empirically quantified fluxes from coral based on water and air measurements (see Results and Discussion). Our calculations are based on few parameters including the net DMS production rate that is also used to infer gross DMS production rate. This approach suggests that the potential for DMS production in coral reefs is very high but much of the climatically important flux of DMS to the atmosphere, where it exerts its cooling activity, is driven by the consumption of DMS through microbial processes. Hence, we use our research to stress the requirement for a better understanding of these consumption processes if we were to improve our forecasting ability of DMS fluxes under ongoing/future environmental change.

We now include reference to the more recent publication by Jones and King (2015). Data from the chamber experiments presented in the paper by Jones et al. (2007) would have been very useful for inclusion in our manuscript. However, as far as we are aware, these experiments were conducted without biological replication, hence lack statistical analysis of the results (e.g. no error presented in their Figure 7) and are presented with confusing (erroneous?) units. Taken together, this precluded us using data from their study as an authoritative reference to enhance our discussion.

END OF RESPONSE TO REFEREE 1

---

## Author Comment (AC2) · 26 Jun 2017

Review of the paper "Quantification of dimethyl sulfide (DMS) production in the sea anemone Aiptasia sp. to simulate the sea-to-air flux from coral reefs" by Filippo Franchini and Micheal Steinke

The paper presented net DMS production and DMSP concentrations in cultures of 48h incubated sea anemones Aiptasia sp. with and without its symbiont Symbiodinium. These data together with literature values were used to estimate the gross DMS production within the anemones and the potential amount of anemone derived DMS emitted to the atmosphere.

This study presents an interesting aspect of the role and influence of sea anemones for the biogeochemical pathways of DMS and DMSP in coral reefs. It shows that even when the production of anemones inside of the polyp is relatively high most of the DMS and DMSP is rapidly consumed and degraded due to microbial activities surrounding the anemones showing again the importance of these sulfur species for the microbial world. Additionally, this study showed that the amount of anemone/coral reef derived DMS for atmospheric processes might be less important than it was thought before suggesting coral reefs as less important hot spots compare with phytoplankton spring blooms in boreal regions. However, the method part of the paper is difficult to understand due to very short descriptions that missing some important details resulting in confusion of the reader. Thus, I suggest publishing this paper after major revision.

Author response (AR) 7: We thank referee 2 for the positive comments and very helpful suggestions for improvements to our manuscript. Following the reviewer's comments, we conducted a major revision of our manuscript that resulted in extensively updated Methods sections (sections 2.5 and 2.6). We also re-analysed our simulation and included confidence intervals for cellular DMSaq for the four Symbiodinium clades in Steinke et al. (2011) in our analysis. This changed the magnitude of the net/gross DMSaq production ratio (R) but not the final outcome of the model. The order of Figures and Tables was changed and added a new Table 2 showing parameters extracted from Steinke et al. (2011).

Major comments The reader gets easily confused by the different terms "net DMS production" and "DMS gross production" and which of the terms are measured or calculated/ estimated. Figure 2 was very helpful to understand but it is mentioned only in the last section of the paper. Please define/specify in your method parts the different terms in one to two sentences and make clear how you determined it.

AR 8: We clarified which parameter was measured or modelled at the end of the intro-duction and at the beginning of Method Section 2.5. Net aqueous DMS production and DMSP concentration within anemones were measured. Gross aqueous DMS produc-tion in anemones and coral-driven sea-to-air DMS flux were simulated.

The anemone gross DMS production calculation is confusing and difficult to understand when it is explained together with the DMS flux calculation in one equation. For a better understanding please explain first the gross DMS production separately and give more information about the different parameter you used in the equation.

AR 9: In Section 2.5 we separated the modelling approach into four steps. (i) Simula-tion of anemone gross DMSaq production rate from measured DMSP and information from the literature (Tables 1 and 2). (ii) Calculation of the ratio (R) between measured net and simulated gross DMSaq production. (iii) Simulation of coral gross DMSaq pro-duction rate. (iv) Conversion of coral gross DMSaq production to coral net DMSaq production using R and subsequently calculation of the sea–to–air flux. More informa-tion for each parameter in the model and an improved explanation of the model are included in the revised version of our manuscript.

It is not completely clear why you chose certain parameters. For instance, why you used DMSP from Yancey et al. 2010 when you have directly measured DMSP and biomass in your incubations?

AR 10: Table 3 was presented to compare the DMSP concentration measured in our study with those in previous studies. Anemone gross DMS production rate was simu-lated from anemone holobiont DMSP values (DMSPH) measured in our study and not those in Yancey et al. 2010. However, because we did not work with corals, coral gross DMS production rate was simulated starting from coral DMSP values (DMSPC) found in the literature (i.e. Yancey at al. 2010, Tab. 1).

Why you chose for NÂËĞnA1, NA2 and so on cell number maximum of 100? Is this a reasonable amount for anemone symbionts in your cultures?

AR 11: The number of Symbiodinium clade cells did not represent the real number within the anemones. It was set arbitrarily to 100 and it randomly changed within the simulation framework in order to generate different community compositions (see updated Section 2.5). Setting N = 1000 did not change the outcome of the model (see caption Table 1).

And please give more information about your previous study Steinke et al. 2011 regarding DMS and Symbiodinium you refer to in this study.

AR 12: This information is now included in the new Table 2.

How did you determined TW?

AR 13: Data for tissue weight (TW) were based on various coral species and taken from Thornhill et al. 2013 (see Table 1). We clarified this in the revised text in section 2.5.

And why is P between 0 and 20 % reasonable for your experiment. Why is the equation for gross DMSaq in anemone the same as the coral gross DMS-production equation (p 3, L 35)? Did you replace the TW for corals in this equation with the TW of the anemone?

AR 14: P is most sensitive to changes in temperature and wind speed and we selected a range of 1 to 20% based on the data presented in Bates et al. (1994). We have re-written the methods section 2.5 including a clearer description of the simulation with two new equations. The new equation 1 describes the calculation of gross DMS production rate in the anemone holobiont, whereas equation 2 describes the calculation of net DMS flux. TW was used to simulate the sea-to-air DMS flux from coral reefs (not included in Eq. 1). Briefly, Eq. 1 was used to simulate the gross DMSaq production rate in anemones using the DMSPH measured in this study and the data in Steinke et al. (2011). The same equation but with DMSPC instead of DMSPH was used to simulate the gross DMS production rate in corals. This was multiplied by TW to convert biomass-normalized coral gross DMS production rates into CSA-normalized coral gross DMS production rates. The resulting values were finally multiplied by R and P to calculate the sea-to-air DMS flux.

Why are the assumptions on p4 L1-7 are reasonable. Please justify. Have you tested it?

AR 15: In the revised version of the manuscript, we explicitly discuss our five assumptions to provide support for our approach (section 3.3). Some of our assumptions are based on few data available in the literature. For example, it is currently impossible to assess whether the ratio between net and gross DMSg production calculated for anemones also applies to corals.

In your experiment, anemones were the organisms of interest, but you talked a lot about corals and coral surface area, so the reader gets confused if you want to show the impact of anemones or corals. You also said "Using our measurements of DMSP concentration and DMS production in anemones to extrapolate to coral reef environments has its limitations..." (p5, L16). Furthermore, on P5 L28: You said that you "normalized to CSA". How did you normalized? Did you assume that anemone coverage in coral reefs was 100% or you assumed that corals and anemones produce similar amounts of DMS so that the composition of the coral reefs (corals or anemones) didn't matter?

AR 16: We are using the anemones (phylum Cnidaria, order Actinaria) as a model system to explore DMS cycling in the globally important coral reefs that are dominated by stony corals (phylum Cnidaria, order Scleractinia). Stony corals are difficult to grow and experiment on. Hence, Aiptasia is often the preferred model to study bleaching and other processes in cnidarians. For example, we would not have been able to conduct a comparison between zooxanthellate and bleached individuals of stony coral species (Fig. 1a), since they have an obligate mutualistic relationship with their endosymbionts. Very little is known about the details of DMS cycling in tropical environments and we

explored the flux of DMS from tropical reefs using the limited published information available in the literature as best as currently possible.

We normalised to CSA by converting biomass-normalised DMS production to surface-normalised DMS production using TW [mg DW cm-2; eq. 2]. We assumed that the DMS production by the endosymbiont Symbiodinium is similar in anemones and corals and that the ratio (R) between net and gross DMS production calculated for anemones (see AR 15) also applies to corals (section 3.3).

Please justify why you can compare anemones and corals and why you can use anemone driven DMS to interpret the amount of DMS produced/released from coral reefs in general. Please say also something about the limitation of this comparison.

AR 17: As suggested by the reviewer, we added this information to the Results and Discussion section 3.3.

In your equation and your Fig 1c, please explain shortly the meaning of the term net DMSaq/gross DMSaq. Does the term say something about the amount of consumed DMS?

AR 18: We added an explanation of this term to the Method section 2.5. Our simulation suggests a ratio of 0.2 suggesting that about 80% of the gross DMSaq is being consumed (likely from reaction with ROS and microbial consumption/catabolism).

The section 2.5 "Data analysis" is very difficult to understand. It needs more details about why and what you were doing with your data. What do you want to say in the first sentence (p4 L10)? Please reword it. Is the mono-factorial analysis well known? Can you shortly say what that mean? What is the R package pse doing, why you used it? The references you gave are very complicated and detailed. It would be great when you give a more general information in your paper. Please, give also a short and general explanation about Monte-Carlo and why you applied it. In the last sentence of section 2.5 (p4 L20-22) is not clear what you have done. Please give more information

how you determined the sensitivity of the variables.

AR 19: We substantially revised the Data Analysis section (section 2.6). The first sentence described how data presented as a column figure in Steinke et al. (2011) were converted into numerical values. In the revised version of our manuscript, we applied original data including error terms in our re-analysis (see AR 7) so that this sentence became obsolete and was removed. Mono-factorial analysis means that the response variable (net DMS production in Fig. 2a) was compared between the two levels ('light' and 'darkness') of the factor 'treatment'. This is principally the same for Fig. 2b but here the factor was the 'production type', i.e. 'net' or 'gross' (2 levels). More information on the pse package and the Monte-Carlo simulation was added to Method Section 2.6 as requested.

Why you didn't determine the net DMSP production? Is this term not interesting?

AR 20: DMSP is a zwitterion and, in contrast to the freely diffusible DMS, does not easily cross cellular membranes. It is likely that observed concentrations of dissolved DMSP (DMSPd) in previous publications are overestimates stemming from the release of DMSP from expelled/non-symbiotic Symbiodinium in the medium (see discussion in Kiene, R. P., and D. Slezak. 2006. Low dissolved DMSP concentrations in seawater revealed by small-volume gravity filtration and dialysis sampling. Limnology and Oceanography-Methods 4: 80-95). Hence, it is difficult to quantify net DMSP production from an accumulation of dissolved DMSP in medium after 48h incubation. Other studies used isotopic labelling coupled with mass-spectrometric detection of DMSP (e.g. Stefels, J., J. W. H. Dacey, and J. T. M. Elzenga. 2009. In vivo DMSP-biosynthesis measurements using stable-isotope incorporation and proton-transfer-reaction mass spectrometry (PTR-MS). Limnol. Oceanogr. Methods 7: 595-611.), techniques that are not available to us. Assuming that anemone were fully acclimated to our experimental set up and growth is negligible during our 48 incubation period, net DMSP production is likely going to be close to zero since the concentration of DMSP per biomass is typically stable at constant environmental conditions.

Minor comments Abstract P1 L10: Please delete the part with the gas chromatograph. I suggest "Here we quantified the net DMS production and the concentration of its cellular precursor dimethylsulfoniopropionate (DMSP) in the cultured sea anemone Aiptasia sp., : : :"

AR 21: Changed as suggested.

Please show only one number after the decimal place in the abstract, e.g. 44.2 instead of 44.22 (p1 L13) and 6 instead of 6.00.

AR 22: Changed as suggested.

P1 L15: This sentence is very confusing. You say that you simulated the DMS flux and than you present the results of the gross DMS production. I suggest "We applied these findings to a Monte-Carlo simulation to demonstrate that net aqueous DMS production accounts for only 0.5 – 2% of gross aqueous DMS production. Monte Carlo based estimations of DMS fluxes into the atmosphere showed that reefs may release up to : : :"

AR 23: Changed as suggested.

Maybe you can write also a discussion sentence about the DMS flux results in the abstract as you have done for DMS gross production.

AR 24: In the abstract we state: '. . .Monte-Carlo based estimations of sea–to–air DMS fluxes showed that reefs may release up to 25 $\mu$mol DMS m-2 coral surface area (CSA) d-1 into the atmosphere with 40% probability for rates between 0.5 and 1.5 $\mu$mol m-2 CSA d-1. These predictions were in agreement with directly quantified fluxes in previous studies. Conversion to a flux normalised to sea surface area (SSA) (range 0.3 to 17.0 with highest probability for 0.3 to 1.0 $\mu$mol DMS m-2 SSA d-1), suggests that coral reefs emit DMS at lower rates than the average global oceanic DMS flux of 6.7 $\mu$mol m-2 SSA d-1 (28.1 Tg sulfur per year). . .'

Section 3.2 You discussed in this section that DMS removal processes under light conditions are faster compared to dark conditions mainly due to microbial consumption. However, in your incubation experiment you didn't see lower DMS concentration in the light treatments compare to the dark treatments. Maybe you should consider and discuss that your incubation experiments didn't contain the microbial diversity as natural environments have. You used artificial seawater (axenic?) for the incubation, thus you might miss important DMS consuming microbes in your experiments resulting in similar DMS concentrations in dark and light treatments.

AR 25: No attempts were made to sterilise the seawater medium. We added information on microbial diversity to our Methods (section 2.1) and a short discussion on the effect of microbial diversity on DMS consumption under Results and Discussion (section 3.2).

P6 L 1: "an average rugosity of 3". Can you say what that means? Is 3 much rugosity or only a little bit? Has rugosity a unit?

AR 26: We added a definition of rugosity to the Results and Discussion (section 3.3).

Fig.1 d: please add the different variables in the figure or color code the dots. It is not clear which point presents which variable in the sensitivity plot. Maybe you can say a little bit about what the different sensitivity numbers mean in the plot, such as "variables close to 0 have less influence on the simulation than variables lower/higher than 0" or something similar.

AR 27: There was an error within the R script in the line coding for the x-axis ticks. We apologise for not having noticed it in the version submitted earlier. Note that we now show the original Figure 1d as a separate Figure 3 in the revised version of our manuscript. We also included the parameters cDMS, CV, and cDMSP for the four Symbiodinium clades that were allowed to vary within the confidence intervals given in Steinke et al. (2011) in our re-analysis. The Figure caption now includes a short description of the sensitivity values: '...Values close to 0 have less influence on the simulation than those departing from 0...'

END OF RESPONSE TO REFEREE 2

---

## Author Response (AR2)

**bg-2017-70: Response to Reviewer's comments**

**"Quantification of dimethyl sulfide (DMS) production in the sea anemone Aiptasia sp. to simulate the sea-to-air flux from coral reefs" by Filippo Franchini and Michael Steinke**

**Reviewer Comment:** The paper has improved enormously. It is now easy to read and to understand. Good job.

**Author response (AR) 1:** *We thank the referee for their comments.*

However, I have some minor comments:

P3 L34: "To quantify the net production of DMS released into the aqueous medium by the holobiont…"; I think you should write "… the net concentration of DMS released…" You cannot release a production into the surrounding water.

**AR2:** *Changed this to: '…To quantify the net production of DMS by the holobiont (net DMSaq production; the release of DMS into the aqueous medium over time)…'*

P4 L20: "Note that N does not reflect the actual number of symbionts within anemones but was arbitrarily set in order to calculate the proportion of clade i among all clades within anemones." How did you decide, which cell numbers are reasonable to use for you simulations. You wrote in table 1 a range of 0-100 cells. That seems to be a quit small number. You said that you tested also with 1000 cells. Are there not millions of Symbiodinium cells in the anemones? Please give a reference or a reason for the small cell numbers you selected and if this reflects the real abundance of symbiont cells in the anemones. Did you run the model also with a higher cell number like 10 000 or even 1 million cells? Is the output still the same?

**AR3:** *We ran the model with 100, 1000 and 1 million cells and the principle outcome is the same. We clarified our approach by making the following change to the text (p.4, l. 20): '…Note that $N$ does not reflect the actual number of symbionts within anemones but was arbitrarily set to 100 in order to calculate the proportion of clade i among all clades within anemones (setting $N$ to $10^3$ or $10^6$ did not change the final outcome of the simulation)…'*

P7 L25-27: "Last but not least corals calcify and this might change the allocation of resources within the host with consequences on the type of relationship with their symbionts

under stress conditions." What do you mean with this? How can a different allocation of resources influence the relationship between host and symbionts? Can you clarify?

*AR4: We agree with the reviewer that this sentence is speculative and not very clear. We decided to remove it from the text to avoid confusion.*

P7 L30: "The percentage of the gross production escaping into the water surrounding the anemones ranged from 1 to 120%..." 120%? Is more released into the water than is produced? This is not realistic.

*AR5: We added the following text to the manuscript to offer an explanation of the high simulated net/gross production rates: '…A percentage of gross DMS production escaping into the atmosphere greater than 100% occurs when the simulated net production exceeds the gross production. This is due to the random sampling of high net-production values (Table 1), and to the calculation of low gross production within the simulation framework. Low gross production arises when a particular combination of parameter values are inserted into Equation 1. For example, a low symbiont population resulting in low $DMSP_H$, combined with a population of low DMS-producing Symbiodinium clades with small cell volume (e.g. Symbiodinium clade B1) could result in this output from the simulation framework. Highest probabilities of 60% were found for net/gross production ratios of 5 to 30% (Fig. 2c)…'*

Fig 3: In the label of x-axis you wrote DMSPA and in the caption you wrote DMSPH. Please change the label. You forgot to describe DMSaq NP in the caption.

*AR6: Corrected as requested by reviewer.*

Ref. Kemp et al. 2014 you wrote "during, and after a coral beaching event ", please change to "bleaching".

*AR7: Corrected as requested by reviewer.*

*Please note that in addition to the changes requested by the reviewer, we also updated one of the in-text references which resulted in the removal of a paper by Lana et al. 2011 and the addition of the paper by Land et al. 2014.*

***END OF RESPONSE***